# β-Boswellic Acid Inhibits RANKL-Induced Osteoclast Differentiation and Function by Attenuating NF-κB and Btk-PLCγ2 Signaling Pathways

**DOI:** 10.3390/molecules26092665

**Published:** 2021-05-01

**Authors:** Gyeong Do Park, Yoon-Hee Cheon, So Young Eun, Chang Hoon Lee, Myeung Su Lee, Ju-Young Kim, Hae Joong Cho

**Affiliations:** 1Musculoskeletal and Immune Disease Research Institute, School of Medicine, Wonkwang University, 460 Iksandae-ro, Iksan, Jeonbuk 54538, Korea; rudeh2508@naver.com (G.D.P.); hanleuni@naver.com (Y.-H.C.); eunsoyg@hanmail.net (S.Y.E.); lch110@wku.ac.kr (C.H.L.); ckhlms@wku.ac.kr (M.S.L.); 2Division of Rheumatology, Department of Internal Medicine, Wonkwang University Hospital, 460 Iksandae-ro, Iksan, Jeonbuk 54538, Korea; 3Department of Obstetrics and Gynecology, Wonkwang University Hospital, Iksan 54538, Korea

**Keywords:** β-boswellic acid, osteoclast, bone resorption, osteoporosis

## Abstract

Osteoporosis is a systemic metabolic bone disorder that is caused by an imbalance in the functions of osteoclasts and osteoblasts and is characterized by excessive bone resorption by osteoclasts. Targeting osteoclast differentiation and bone resorption is considered a good fundamental solution for overcoming bone diseases. β-boswellic acid (βBA) is a natural compound found in *Boswellia serrata*, which is an active ingredient with anti-inflammatory, anti-rheumatic, and anti-cancer effects. Here, we explored the anti-resorptive effect of βBA on osteoclastogenesis. βBA significantly inhibited the formation of tartrate-resistant acid phosphatase-positive osteoclasts induced by receptor activator of nuclear factor-B ligand (RANKL) and suppressed bone resorption without any cytotoxicity. Interestingly, βBA significantly inhibited the phosphorylation of IκB, Btk, and PLCγ2 and the degradation of IκB. Additionally, βBA strongly inhibited the mRNA and protein expression of c-Fos and NFATc1 induced by RANKL and subsequently attenuated the expression of osteoclast marker genes, such as *OC-STAMP, DC-STAMP, β3-integrin, MMP9, ATP6v0d2,* and *CtsK*. These results suggest that βBA is a potential therapeutic candidate for the treatment of excessive osteoclast-induced bone diseases such as osteoporosis.

## 1. Introduction

Bone is constantly formed by osteoblasts and resorbed by osteoclasts throughout its lifetime. Bone remodeling is a tight coupling process involving crosstalk between osteoblasts and osteoclasts, which is crucial for maintaining normal bone mass and systemic mineral homeostasis [1,2,3]. Osteoclasts are multinucleated cells derived from hematopoietic precursor cells that are stimulated by macrophage colony-stimulating factor (M-CSF) and receptor activator of nuclear factor κ-B (RANKL). Excessive activity of osteoclasts may cause a wide range of osteolytic diseases such as osteoporosis, periodontitis, and rheumatoid arthritis; thus, the control of osteoclast activity and metabolic pathways is crucial in the treatment of such diseases [1,2,3,4,5]. Successful therapeutic strategies for treatments of osteolytic diseases include bisphosphonate and denosumab. However, long-term use of these drugs can cause serious side effects such as atraumatic bone fracture, increased esophageal cancer, atypical fractures, and bone necrosis [6]. In addition, denosumab has been reported to have infectious diseases and various side effects that increase bone or muscle pain [7]. Thus, the side effects of osteolytic diseases treatments have led to an interest in research into the discovery for the prevention of bone diseases from plant-based products or other effective extracts [8].

RANKL, a member of the TNF receptor family, acts as an essential regulator of osteoclast differentiation and activation. The process depends on the RANK–RANKL interaction, which simultaneously regulates various adaptor molecules, such as TNF receptor-associated factors (TRAFs). Among the TRAFs, TRAF6 is a crucial downstream mediator of RANKL–RANK signaling, and TRAF6 binds to RANK to form a trimer that triggers numerous downstream signals, such as the IκB kinase (IKK) complex to stimulate nuclear factor kappa B (NF-κB) [9] and MAPKs, including p38, ERK, and JNK [10,11,12]. The adaptor function of PLCγ2 is required for activator protein-1 (AP-1) and NF-κB [13]. RANKL-TRAF6-mediated signaling ultimately activates nuclear factor of activated T cells 1 (NFATc1), which acts as a master transcription factor for the terminal differentiation of osteoclasts, and regulates the expression of various osteoclast-specific genes, such as *osteoclast stimulatory transmembrane protein* (*OC-STAMP*) [14], *dendritic cell-specific transmembrane protein* (*DC-STAMP*) [15], *ATPase H^+^-transporting V0 subunit d2* (*ATP6v0d2*), and *cathepsin K* (*CtsK*) [16,17]. Therefore, inhibition of RANKL-induced NFATc1 signaling is important for the development of novel therapeutic approaches against bone diseases. 

Traditionally, natural products have been used as important, safe, and convenient resources in medicines against cancer, inflammation, oxidant activity, immunosuppression, and allergies [18]. Boswellic acids (BAs) are a series of pentacyclic terpenoid molecules produced by *Boswellia serrata* that have anti-inflammatory, anti-rheumatic, anti-asthmatic, and anti-cancer effects. There are more than 12 types of BAs of which α-boswellic acid (αBA), β-boswellic acid (βBA), keto β-boswellic acid (KBA), and acetyl-keto β-boswellic acid (AKBA) are the major ones [19]. Previous studies reported that AKBA inhibits osteoclastogenesis by suppressing the NF-κB-induced TNF-α signaling pathway [20] and attenuates titanium particle-induced osteogenic inhibition via the GSK-3β/β-catenin signaling pathway [21]. However, the biological effects of βBA on osteoclast differentiation and bone resorption remain unknown. Therefore, we investigated the effect of βBA on RANKL-induced osteoclastogenesis and elucidated its regulatory mechanism.

In this study, we found that βBA inhibits osteoclast formation and bone resorption and downregulates NFATc1 expression via suppression of NF-κB and PLCγ2 signaling in vitro.

## 2. Results

### 2.1. βBA Suppresses RANKL-Induced Osteoclastogenesis without Cytotoxicity

To determine the effect of βBA on osteoclast differentiation, bone marrow-derived macrophages (BMMs) were cultured with M-CSF (30 ng/mL) and RANKL (100 ng/mL) in the presence or absence of various concentrations of βBA (0, 5, 10, 20, and 30 μM) for 3 days. βBA treatment decreased the number and size of TRAP^+^ osteoclasts in a dose-dependent manner (Figure 1B,C). To explore whether βBA induces cytotoxicity during RANKL-induced osteoclast differentiation, BMMs were cultured with M-CSF (30 ng/mL) with various concentrations of βBA (0, 5, 10, 20, and 30 μM) for 3 days. The viability of cells treated with βBA had no toxic effects at any of the concentrations used in the present study (Figure 1A).

### 2.2. βBA Inhibits Bone Resorption Activity 

Next, we examined whether βBA affected bone resorption by mature osteoclasts. The mature osteoclast cells were formed by co-culture with primary osteoblasts (pOBs), and isolated mature osteoclasts were transferred to hydroxyapatite plates and dentin slices and cultured with or without βBA. The survival of TRAP^+^ osteoclast cells did not differ between the control and βBA groups. However, the resorption area was significantly decreased in both the hydroxyapatite plate and dentin slices in βBA compared to the control (Figure 2). These data suggest that βBA suppresses bone resorption without inducing cell death.

### 2.3. βBA Suppresses RANKL-Stimulated Activation of NF-κB and Btk-PLCγ2 Signaling Pathway

As mentioned in many studies, the binding of RANKL to RANK activates various downstream signaling pathways, including TRAF6, NF-κB transcription factor, Akt, MAPKs [10,11,12], and PLCγ2 [13]. To explore the specific pathway in RANKL signaling regulated by βBA, BMMs were serum-starved for 3 h, pretreated with vehicle or βBA (30 μM) for 1 h, and then stimulated with RANKL for the indicated times (0, 5, 15, and 30 min). The effects of βBA on RANKL-stimulated early signal transduction, including Akt, p38 MAPK, NF-κB, and Btk-PLCγ2 activations, were evaluated by immunoblotting. The results showed that the relative expression of phosphorylated Akt and p38 was unaffected in the βBA-treated group compared to that in the control group. However, the expression of phosphorylated IκB was significantly reduced compared with the control. The further degradation of IκB was downregulated, and immunoblotting revealed an increase in the expression band and intensity of IκB. Additionally, βBA negatively affected the phosphorylation of Btk and PLCγ2, which are crucial for the efficient activation of calcium signaling during osteoclast differentiation (Figure 3A,B). These results indicate that βBA regulates NF-κB activation by inhibiting the phosphorylation and degradation of IκB and controls calcium signaling by suppressing the phosphorylation of Btk and PLCγ2 in osteoclast differentiation.

### 2.4. βBA Downregulates the Expression Levels of c-Fos and NFATc1 Transcription Factor in Osteoclast Differentiation 

To further study the mechanism of the inhibitory effects of βBA on RANKL-induced osteoclast differentiation and resorption function, we evaluated the effects of βBA on RANKL-induced c-Fos and NFATc1 expression using real-time RT-PCR and immunoblotting. As expected, RANKL stimulation significantly induced the mRNA expression of *c-Fos* and *NFATc1*. RANKL-induced *c-Fos* and *NFATc1* mRNA levels were significantly reduced by βBA treatment (Figure 4A). Additionally, immunoblotting demonstrated that βBA dramatically reduced the protein levels of c-Fos and NFATc1 (Figure 4B). These results suggest that βBA inhibits osteoclast differentiation by reducing the expression levels of c-Fos and NFATc1 stimulated by RANKL.

### 2.5. βBA Suppresses the mRNA Expression of Osteoclast Specific Marker Genes

During osteoclastogenesis, the transcription factor NFATc1 regulates the transcription of osteoclast-specific marker genes, including *OC-STAMP, DC-STAMP, β3-integrin, matrix metallopeptidase 9 (MMP9), Atp6v0d2,* and *CtsK*. To investigate whether βBA exhibits anti-osteoclastogenic effects by regulating the transcription of marker genes by NFATc1, the mRNA expression of marker genes was confirmed by real-time RT-PCR. βBA significantly inhibited the mRNA expression of *OC-STAMP* and *DC-STAMP**,* which are crucial osteoclast fusion-related molecules. The expression of *β3-integrin* and *MMP9*, which are essential for osteoclast resorption, was strongly suppressed by βBA treatment. Finally, βBA significantly attenuated the mRNA expression of *ATP6v0d2* and *CtsK*, which are essential components related to extracellular acidification and maintenance of bone resorption. These results suggest that βBA strongly downregulates osteoclastogenic marker genes related to cell-to-cell fusion and bone resorption during osteoclastogenesis. 

## 3. Discussion

Extracts of *B. serrata* have been used in the Ayurvedic system (India’s traditional medicine) for relieving numerous inflammatory disorders or infections, such as bowel disease, rheumatoid arthritis, osteoarthritis, chronic pain syndrome, and asthma. Various BA compounds have been identified as effective bioactive molecules in *Boswellia* species. However, pharmacokinetic studies have shown low systemic absorption of BAs, especially of KBA and AKBA, in rodents and humans. Hüsch et al. [22] measured the concentrations of the six major BAs (KBA, AKBA, βBA, AβBA, αBA, and AαBA) at the plasma level using soy lectin formulation with BAs to improve the formulation of BAs. In particular, βBA (3-fold) and KBA (7-fold) only revealed acceptable plasma levels, since their absorption efficiency is much higher than that of AKBA, AβBA, αBA, and AαBA in vivo [22]. Another permeability experiment using BAs showed that the plasma concentration of BA was 100-fold higher than that of AKBA and KBA. βBA may play an important role by targeting microsomal prostaglandin (PG) E2 synthase-1 and cathepsin G. However, AKBA had no significant effects in these experiments [23]. βBA also suppressed pleurisy in a carrageenan-induced pleurisy rat model by intraperitoneal and oral administration of a low dose of βBA (1 mg/g) [24]. These reports indicate that βBA has a good absorption rate in vivo at a lower concentration than other isoforms. Therefore, βBA may be more effective as a therapeutic agent against diseases such as immune-related disorders [24,25,26].

Henkel et al. suggested that βBA acts as a direct LPS inhibitor and may contribute to anti-inflammatory effects by inhibiting LPS activity in a range of blood plasma levels (6.4–10.1 μM) after oral administration of standard doses of BA extract [22,27,28]. The neutralization mechanism of LPS by βBA may be a key part of the molecular actions responsible for valuable BA effects and might be one of the significant BA derivatives in LPS-induced inflammatory diseases. Several studies on BA compounds have also been reported on AKBA in bone-related diseases. Orally administered AKBA exerts anti-arthritic activity in bovine serum albumin-induced arthritis. Additionally, AKBA inhibits the toll-like receptor-mediated activation of monocytes through the downregulation of LPS-induced nitric oxide, IL-β, and TNF-α production [18,19,20,21]. Therefore, this study used βBA to develop a therapeutic agent against bone diseases, which has higher absorption efficiency in vivo than AKBA, may be effective as a drug. 

RANKL–RANK interaction activates the trimerization of RANK and recruits TRAF6 adaptor protein, which is a crucial downstream mediator. TRAF6, in turn, binds with RANK and activates the sequential phosphorylation of MKK3/6, p38 MAPK, and ATF2, inducing osteoclast differentiation. Furthermore, TRAF6 initiates TANK-binding kinase 1 belonging to the IκB kinase (IKK) family and activates PI3K/Akt activation depending on Src kinase activity [29]. In this study, βBA did not affect Akt or p38 signaling (Figure 3A,B). RANKL stimulation also triggered the activation of NF-κB, AP-1, and PLCγ2. TRAF6 interacts with Grb-2-associated binder-2 and PLCγ2 as an adaptor molecule for RANK. NF-κB is a crucial transcription factor in osteoclast differentiation. NF-κB signaling compromises several activation steps that require ubiquitination and proteasome degradation or processing of proteins that function as inhibitors, including the canonical IκBs (IκBα, IκBβ, and IκBε). The IκBs interfere with the function of NF-κB protein by binding to NF-κB dimers to inhibit their nuclear localization. The NF-κB family consists of p65, p50, p52, RelB, and Rel. The NF-κB canonical signaling is mediated by RelA/p50 in response to stimuli such as RANKL, RNF, and IL-1. The non-canonical NF-κB signaling is initiated by the translocation of RelB/52 heterodimers. After RANKL stimulation, IKKβ is activated and initiates the ubiquitin-mediated proteasome degradation of IκB, which eventually results in the release of NF-κB from the IκB–NF-κB complex [30]. PLCγ2 is critical for the activation of the calcium signaling pathway in osteoclast differentiation. NFATc1 activation was impaired by PLCγ2 deficiency. The activation of PLCγ2 by RANKL stimuli is positively regulated by Btk activation. The auto-amplification of NFATc1 expression depends on continuous Btk-PLCγ2 signaling activation [31]. In this study, βBA suppressed osteoclast differentiation via reduced nuclear translocation of NF-κB by inhibiting IκB degradation and Btk-PLCγ2 calcium signaling (Figure 3A). 

NFATc1, a key transcription factor in osteoclast differentiation, regulates the transcription of osteoclast marker genes. During the later stages of osteoclast maturation, NFATc1 regulates the expression of OC-STAMP, and DC-STAMP regulates the cell-to-cell fusion of osteoclasts. The β3-integrin cytoplasmic domain interacts with c-Src and c-Cbl, which is responsible for the activation of the adhesive complex of osteoclasts [14,15,16]. MMP9, a type IV collagenase that belongs to the zinc-binding enzyme family, is highly expressed in osteoclasts. MMP-9 plays an important role in extracellular matrix degradation and is responsible for bone resorption and bone remodeling [32]. Atp6v0d2 acts as an extracellular acidification regulator and affects osteoclast maturation [16]. CtsK is a lysosomal cysteine proteinase responsible for bone resorption of bone matrix through regulation of proteolytic activity [33]. In this study, βBA inhibited osteoclast differentiation by inhibiting the expression of c-Fos and NFATc1 transcription factors (Figure 4A,B) and reduced the mRNA expression of marker genes such as *OC-STAMP*, *DC-STAMP*, *β3-integrin*, *MMP9*, *ATP6v0d2*, and *CtsK* (Figure 5). Therefore, βBA inhibited the bone resorption of osteoclasts (Figure 2). 

Taken together, these results suggest that βBA inhibits NFATc1 transcription during RANKL-induced osteoclast differentiation and prevents osteoclast maturation and bone resorption by blocking NF-κB and Btk-PLCγ2 signaling. However, we could not confirm the effect of βBA on osteoblast and animal study. Therefore, further research on the effect of βBA on osteoblasts could provide more insight into the effect of inhibiting osteoclastogenesis, and the efficacy of βBA in vivo could suggest therapeutic potential. 

## 4. Materials and Methods

### 4.1. Chemicals, Reagents, and Antibodies

βBA, obtained from Sigma-Aldrich (St. Louis, MO, USA), was dissolved and prepared with 5, 10, 20, and 30 mM stock solutions in DMSO. The control group was added 0.1% DMSO. TRIzol reagent was obtained from Life Technologies (Carlsbad, CA, USA). A monoclonal anti-β-actin antibody was obtained from Sigma-Aldrich (St. Louis, MO, USA). Antibodies against phospho-Akt, anti-Akt, anti-phospho-p38, anti-total p38, anti-phospho-IκB, anti-Bruton’s tyrosine kinase, and anti-phospho-PLCγ2 were obtained from Cell Signaling Technology Inc. (Beverly, MA, USA). Anti-phospho-Btk antibody was obtained from GeneTex (Irvine, CA, USA). Anti-c-Fos, anti-NFATc1, anti-IκB, and anti-PLCγ2 antibodies were purchased from Santa Cruz Biotechnology (Santa Cruz, CA, USA). Donkey anti-rabbit and anti-mouse immunoglobulin secondary antibodies were purchased from Enzo Life Sciences (Farmingdale, NY, USA).

### 4.2. Primary BMM and Osteoblasts (OBs) Cells Culture System

BMMs and osteoblast cells were prepared as previously described [24,25,26]. All experiments related to primary cell culture using mice were conducted according to the guidelines of the Institutional Animal Care and Use Committee (IACUC) of Wonkwang University (Approval number: WKU20-39). Bone marrow cells (BMCs) were prepared from 5-week-old male ICR mice by flushing the femurs and tibias with α-MEM supplemented with 1% penicillin/streptomycin (P/S) antibiotics. BMCs were cultured on 100 mm culture dishes in α-MEM supplemented with 10% FBS and M-CSF (10 ng/mL) for 1 day. Non-adherent cells were plated in 90 mm Petri dishes and cultured in α-MEM supplemented with 10% FBS, 1% P/S, and M-CSF (30 ng/mL). After the non-adherent cells were removed, the adherent cells were used as BMMs. Primary OBs were prepared with the calvaria of neonatal 1 day mice and were digested five times each 15 min with 0.1% collagenase I (Life Technologies, Grand Island, NY, USA) and 0.2% neutral protease dispase II (Roche, Basel, Switzerland) in α-MEM supplemented with 1% P/S. The cells further cultured and isolated in the α-MEM supplemented with 10% FBS and 1% P/S for 3 days. 

### 4.3. BMM Cytotoxicity Assay

BMMs (1 × 10^4^ cells) were cultured in α-MEM supplemented with 10% FBS and M-CSF (30 ng/mL) at various concentrations of βBA (0, 5, 10, 20, and 30 μM) in a 96-well plate and incubated for 3 days. Thereafter, the cell viability was analyzed by adding 50 μL of tetrazolium salt sodium 3′-{1-[phenylamino]-carbonyl}-3,4-tetrazolium}-bis (4-methoxy-6-nitro) benzene-sulfonic acid hydrate (XTT reagent) to each well, followed by another incubation for 4 h. Absorbance was measured at 450 nm using a multi-detection microplate reader (Molecular Devices, Sunnyvale, CA, USA). 

### 4.4. In Vitro Osteoclastogenesis Assay 

To induce osteoclast differentiation using BMMs, the cells were seeded in a 48-well plate (3.5 × 10^4^ cells/well) in α-MEM supplemented with 10% FBS containing M-CSF (30 ng/mL) and RANKL (100 ng/mL) and cultured for 4 days. Osteoclasts were identified by TRAP staining. BMM-derived osteoclast MNCs were fixed in 3.7% formalin for 15 min, permeabilized with 0.1% Triton X-100 for 10 min, and then stained with TRAP (Sigma, St. Louis, MO, USA). TRAP^+^ cells containing more than three nuclei were counted as osteoclasts.

### 4.5. Bone Resorption Assay 

Mature OCs were prepared from a co-culture of BMCs and primary OBs. Briefly, BMCs (1 × 10^7^ cells) and primary OBs (5 × 10^5^ cells) were incubated in collagen gel-coated culture dishes in the presence of 10^−8^ M VitD3 and 10^−6^ M PGE2 for 11 days. Mature OCs were detached using 0.1% collagenase and re-seeded in hydroxyapatite plates and dentin slices. The cells seeded on the hydroxyapatite plate and dentin slices were cultured for 24 and 48 h, respectively, and were completely removed. The resorption pits were determined under a microscope and quantified using Image-Pro Plus version 4.5 (Media Cybernetics, Silver Spring, MD, USA). 

### 4.6. Immunoblotting

After treatment and stimulation, cells were rinsed three times with cold phosphate-buffered saline (PBS) and lysed with lysis buffer containing 50 mM Tris-HCl, 150 mM NaCl, 5 mM EDTA, 1% Triton X-100, 1 mM sodium fluoride, 1 mM sodium vanadate, 1% deoxycholate, and protease inhibitor mixture. Lysates were cleared by centrifugation at 13,500 rpm for 20 min at 4 °C, and the supernatants were collected. Protein concentrations were determined using the Bio-Rad DC Protein Assay Kit (Bio-Rad Laboratories Inc., Hercules, CA, USA). For immunoblotting, equivalent amounts of protein (20–30 μg) were resolved by 10% sodium dodecyl sulfate polyacrylamide gel electrophoresis (SDS-PAGE) and transferred to polyvinylidene difluoride membranes (Millipore, Bedford, MA, USA). To block non-specific binding, membranes were incubated with 5% (*w/v*) skim milk in Tris-buffered saline with 0.1% Tween 20 (TBST) for 1 h, washed with TBST, and incubated with primary antibodies overnight at 4 °C. Membranes were washed and incubated for 90 min with the appropriate secondary antibodies. After washing three times with TBST, the target band was detected using Immobilon Western Chemiluminescent HRP Substrate (Millipore, Billerica, MA, USA). 

### 4.7. Real-Time RT-PCR

First-strand cDNA was synthesized from 1 μg of total RNA using SuperScript II Reverse Transcriptase purchased from Thermo Scientific (Thermo Fisher Scientific, Wilmington, DE, USA). The PCR reaction with the cDNA template was performed with Accupower Green Star qRT-PCR master mix using an ExcyclerTM 96 Real-Time Quantitative Thermal Block (Bioneer, Daejeon, Korea). The following sequences of mouse primers were used for real-time RT-PCR: *GAPDH* Forward: 5′-TCAAGAAGGTGGTGAAGCAG-3′; reverse: 5′-AGTGGGAGTTGCTGTTGAAGT-3′; *c-Fos* forward: 5′-AGTCCATTTGCTGACCCCAC-3′; reverse: 5′-GGATGGTCGTGTTGATGCG-3′; *NFATc1* forward: 5′-GAGTACACCTTCCAGCACCTT-3′; reverse: 5′-TATGATGTCGGGGAAAGAGA-3′; *OC-STAMP* forward: 5′-ATGAGGACCATCAGGGCAGCCACG-3′; reverse: 5′-GGAGAAGCTGGGTCAGTAGTTCGT-3′; *DC-STAMP* forward: 5′-TCCTCCATGAACAAACAGTTCCA-3′; reverse: 5′-AGACGTGGTTTAGGAATGCAGCTC-3′; *Atp6v0d2* forward: 5′-GACCCTGTGGCACTTTTTGT-3′; reverse: 5′-GTGTTTGAGCTTGGGGAGAA-3′; *CtsK* forward: 5′-CCAGTGGGAGCTATGGAAGA-3′; reverse: 5′-CTCCAGGTTATGGGCAGAGA-3′. The amplification parameters were as follows: initial denaturation at 95 °C for 5 min, followed by 40 cycles of 3-step PCR: denaturation at 95 °C for 1 min, annealing at 60 °C for 30 s, and a final extension at 72 °C for 1 min. The expression levels of mRNA were calculated as fold changes by evaluating real-time RT-PCR data using the 2^−ΔΔCt^ method. The data were normalized to the mRNA levels of *GAPDH*. 

### 4.8. Statistical Analysis

All experiments were performed at least three times, and data are presented as the mean ± standard deviation (SD). Statistical differences were confirmed using one-way or repeated-measures ANOVA followed by Tukey’s HSD test. Statistical significance was set at *p* < 0.05.

## 5. Conclusions

In conclusion, our results suggest that βBA exerts inhibitory effects on RANKL-induced osteoclast differentiation and bone-resorbing activity and cell fusion in vitro. Taken together, βBA is suggested as a potential therapeutic target for the treatment of bone loss. 

## Figures and Tables

**Figure 1 molecules-26-02665-f001:**
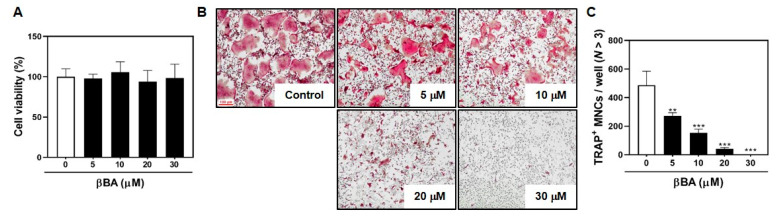
β-boswellic acid (βBA) inhibits RANKL-induced osteoclast differentiation without cytotoxicity. (**A**) Mouse bone marrow-derived macrophages (BMMs) were cultured for 3 days at the indicated concentration of βBA in the presence of M-CSF (30 ng/mL). Cell viability was analyzed by XTT assay. (**B**) BMMs were cultured for 4 days in the presence of M-CSF (30 ng/mL) and RANKL (100 ng/mL) with the control (DMSO) or βBA at the indicated concentration. After culturing, cells were fixed with 3.7% formalin, permeabilized with 0.1% Triton X-100, and stained with TRAP staining solution. (**C**) TRAP-positive multinucleated cells (TRAP^+^ MNCs; *N* > 3) were counted as osteoclasts. Scale bar, 100 μm. Data represent means ± SD (n = 3). ** *p* < 0.01, *** *p* < 0.001 as compared with DMSO control.

**Figure 2 molecules-26-02665-f002:**
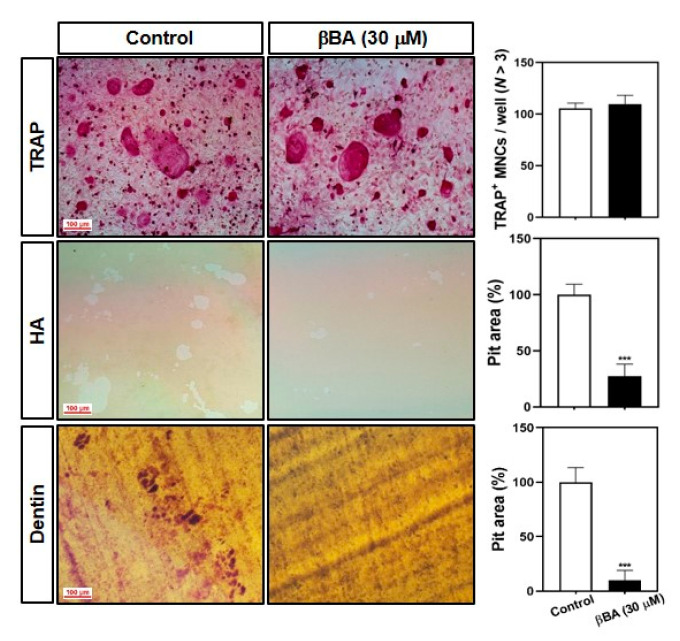
β-boswellic acid (βBA) suppresses the bone-resorbing activity of mature osteoclasts. Mature osteoclasts were prepared with co-cultured osteoblasts and bone marrow cells on a collagen matrix plate. Mature osteoclasts were cultured in a 48-well plate for 24 h, in a hydroxyapatite-coated plate (HA) for 24 h, or in dentin slices for 48 h with or without βBA (30 μM). The cells attached to 48-well plates were stained with a TRAP solution (top), and the cells on HA (middle) and dentin slices (bottom) were removed with 10% bleach solution. The dentin slices were counterstained with hematoxylin solution, and the resorbed surface was photographed under a light microscope. The number of TRAP-positive multinucleated cells (TRAP^+^ MNCs; *N* > 3) was counted, and the relative pit areas in the HA and dentin slices were quantified using Image-Pro Plus (Ver 4.5) software. Scale bar, 100 μm. Data represent means ± SD (n = 3). *** *p* < 0.001 vs. as compared with DMSO control.

**Figure 3 molecules-26-02665-f003:**
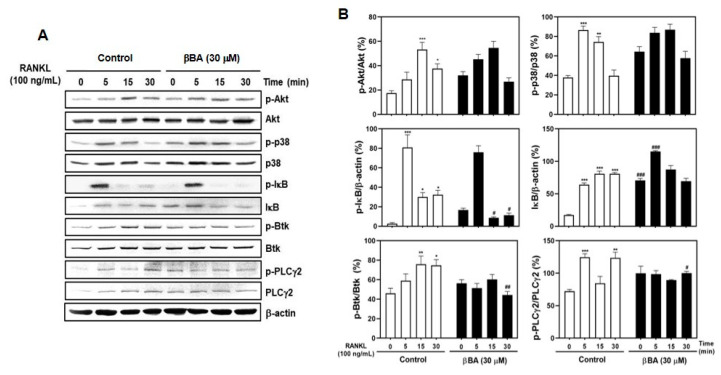
β-boswellic acid (βBA) downregulates RANKL-induced phosphorylation of IκB, Btk, and PLCγ2 and IκB degradation. Mouse bone marrow-derived macrophages (BMMs) were starved with serum-free α-MEM media for 3 h. BMMs were pre-treated with DMSO control or βBA (30 μM) for 1 h and then stimulated with RANKL (100 ng/mL) at the indicated times. (**A**) The cell lysates were analyzed by Western blotting with antibodies. β-actin was used as internal control. (**B**) Quantification of relative ratio of band intensity was performed using Image J software. Data represent means ± SD (n = 3). * *p* < 0.05, ** *p* < 0.01, *** *p* < 0.001 vs. control at 0 min; ^#^
*p* < 0.05, ^##^
*p* < 0.01, ^###^
*p* < 0. as compared with DMSO control at the indicated time.

**Figure 4 molecules-26-02665-f004:**
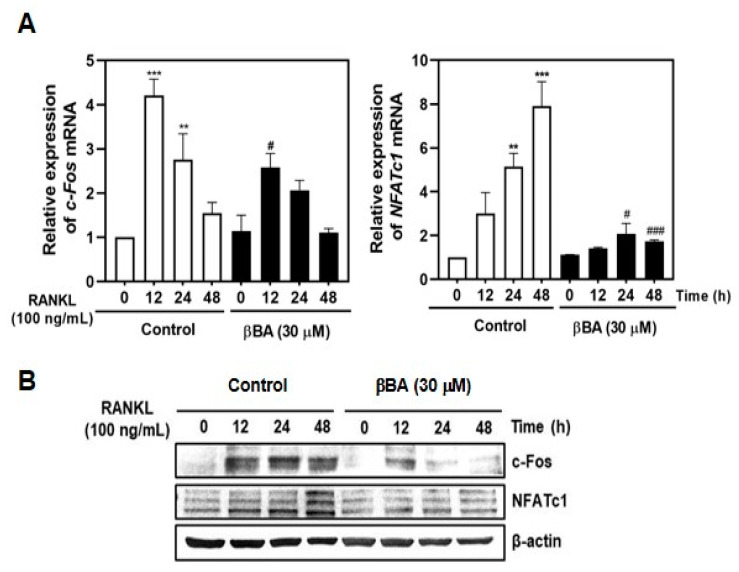
β-boswellic acid (βBA) inhibits RANKL-induced c-Fos and NFATc1 expression. Mouse bone marrow-derived macrophages (BMMs) were incubated with M-CSF (30 ng/mL) and RANKL (100 ng/mL) in the presence or absence of βBA (30 μg/mL) for the indicated time. DMSO was used as control. (**A**) Total RNA was isolated from cells using TRIzol reagent, and mRNA expression levels of *c-Fos* and *NFATc1* were analyzed by real-time RT-PCR. Data represent means ± SD (n = 3). ** *p* < 0.01, *** *p* < 0.001 vs. control at 0 h; ^#^
*p* < 0.05, ^###^
*p* < 0.001 as compared with DMSO control at the indicated time. (**B**) Protein expression levels of c-Fos and NFATc1 were evaluated by immunoblotting. β-actin was used as the internal control.

**Figure 5 molecules-26-02665-f005:**
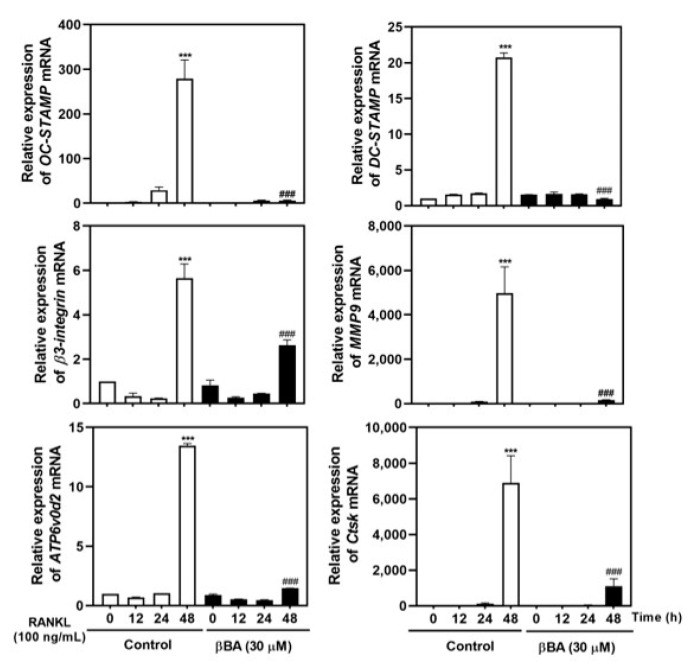
β-boswellic acid (βBA) inhibits RANKL-induced mRNA expression of *OC-STAMP*, *DC-STAMP*, *β3-integrin*, *MMP9*, *ATP6v0d2*, and *CtsK*. Mouse bone marrow-derived macrophages (BMMs) were incubated with M-CSF (30 ng/mL) and RANKL (100 ng/mL) in the presence or absence of βBA (30 μM) for the indicated time. Total RNA was isolated from cells using TRIzol reagent, and the mRNA expression of *OC-STAMP*, *DC-STAMP*, *β3-integrin*, *MMP9*, *ATP6v0d2*, and *CtsK* was analyzed by real-time RT-PCR. Data represent means ± SD (n = 4). *** *p* < 0.001 vs. control at 0 h; ^###^
*p* < 0.001 as compared with DMSO control at the indicated time.

## Data Availability

The data presented in this study are available in the manuscript.

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
