# Peer review of "β-Boswellic Acid Inhibits RANKL-Induced Osteoclast Differentiation and Function by Attenuating NF-κB and Btk-PLCγ2 Signaling Pathways"

_molecules, 2021, doi:10.3390/molecules26092665_

Round 1
Reviewer 1 Report
The aim of this report is to clarify the effects of beta-BA on osteoclast differentiation and resorbing function in BMMs. Experimental designs are well considered, and results are clearly described.
Please see my comments below:
Overall
The authors analyzed and discussed the effects of beta-BA on osteoclast differentiation, including signaling pathways. In addition, Fig. 2 shows the possibility that exposure to beta-BA after the formation of mature osteoclasts also suppresses their function. The reviewer consider that additional discussion is needed as no mention is made of the mechanism by which beta-BA is involved in the function of mature osteoclasts.
Comments
- Why did the author expose beta-BA for 3 days in the cytotoxicity assay; the cells for TRAP staining were exposed beta-BA for 4 days, thus experiments with the same duration would be required in order to analyze accurate cytotoxicity.
- Why was only IkappaB quantified using beta-actin instead of phosphorylated protein?
- In materials and methods section, recommend reordering the headings to match the order of the results.
- Describe the approval number of animal experiments approved by the authors’ institute.
- Describe the preparing method of primary OBs in more detail.
- Describe the number of samples used in each experiment.
- Insert scale bars to representative photos in each figure.
Reviewer 2 Report
The authors investigated the anti-resorptive effect of a natural compound called β-boswellic acid (βBA) identified from Boswellia serrata. The results indicated that βBA inhibited the formation of tartrate-resistant acid phosphatase-positive osteoclasts induced by receptor activator of nuclear factor-B ligand (RANKL) and suppressed bone resorption without any cytotoxicity. The authors also claimed that βBA may be a potential therapeutic candidate for the treatment of excessive osteoclast-induced bone diseases such as osteoporosis.
Some minor concerns are existed before accepting to publish.
- For the introduction section, the authors pointed out that excessive activity of osteoclasts may cause osteolytic diseases, and thus, it is important to control those activities. However, more background information is desired. Which diseases are most affected by these activities? What is the current management for these conditions? How are the treatment effects of the current intervention and do they have any side effects? The authors should provide a more concrete rationale for this study. If the current treatment has well fulfilled the treatment needed without any side effects, why are we still developing new drugs for the conditions?
- For sections 2.2 to 2.5, what is the control for those experiments? The authors should add it to the method section. Additionally, how did the authors determine the treatment dosage of βBA? Will there be dose-dependent with the treatment of βBA? Also, I am wondering why the author did not include a positive control for these experiments?
- For the control group in Figure 5, it is quite weird the expression of OC-STAMP, β3-integrin and ATP6v0d2 mRNA was dramatically increased at 48 hours. I have my doubts about these data.
- For the discussion section, limitations of this study should be discussed as well as future directions.
Round 2
Reviewer 1 Report
I believe that the authors have responded appropriately to my points.